Quantifying cryptic Symbiodinium diversity within Orbicella faveolata and Orbicella franksi at the Flower Garden Banks, Gulf of Mexico

Green Elizabeth A. 1
Davies Sarah W. 2
Matz Mikhail V. 2
Medina Mónica 3 monicamedina@psu.edu
1 Quantitative and Systems Biology, University of California , Merced, CA , USA
2 Department of Integrative Biology, The University of Texas at Austin , TX , USA
3 Department of Biology, Pennsylvania State University , University Park, PA , USA
Thompson Fabiano
Electronic publication date: 2014 May 13
Publication date: 2014
Volume: 2
Electronic Location ID: e386
Received 2014 Feb 23; Accepted 2014 Apr 28
Copyright: © 2014 Green et al.
Copyright year: 2014
Copyright holder: Green et al.
License: This is an open access article distributed under the terms of the Creative Commons Attribution License, which permits unrestricted use, distribution, reproduction and adaptation in any medium and for any purpose provided that it is properly attributed. For attribution, the original author(s), title, publication source (PeerJ) and either DOI or URL of the article must be cited.
License URL: https://creativecommons.org/licenses/by/4.0/

Keywords: Next-generation sequencing (NGS), Flower Garden Banks, Symbiodinium, Caribbean, Orbicella faveolata, Orbicella franksi, ITS-2, OTU, Metabarcoding, Linear mixed model

Funding: NSF DEB-1054766 IOS 0644438 0926906 PADI Foundation Award Research was funded by the National Science Foundation grant DEB-1054766 to MVM and IOS 0644438 and IOS 0926906 to Mónica Medina, and the PADI Foundation Award to SWD. The funders had no role in study design, data collection and analysis, decision to publish, or preparation of the manuscript.

==============================
The genetic composition of the resident Symbiodinium endosymbionts can strongly modulate the physiological performance of reef-building corals. Here, we used quantitative metabarcoding to investigate Symbiodinium genetic diversity in two species of mountainous star corals, Orbicella franksi and Orbicella faveolata, from two reefs separated by 19 km of deep water. We aimed to determine if the frequency of different symbiont genotypes varied with respect to coral host species or geographic location. Our results demonstrate that across the two reefs both coral species contained seven haplotypes of Symbiodinium, all identifiable as clade B and most closely related to type B1. Five of these haplotypes have not been previously described and may be endemic to the Flower Garden Banks. No significant differences in symbiont composition were detected between the two coral species. However, significant quantitative differences were detected between the east and west banks for three background haplotypes comprising 0.1%–10% of the total. The quantitative metabarcoding approach described here can help to sensitively characterize cryptic genetic diversity of Symbiodinium and potentially contribute to the understanding of physiological variations among coral populations.

Introduction

The symbiotic relationship between scleractinian corals and dinoflagellate algae in the genus Symbiodinium is well known, but there is still much to understand about the establishment and plasticity of this complex symbiosis. Knowledge of Symbiodinium taxonomic diversity has increased over the last two decades with advancing molecular genotyping techniques detecting novel genotypes within each of the nine recognized clades (Coffroth & Santos, 2005; Pochon & Gates, 2010). Some of these genotypes may impart different physiological benefits and evidence suggests that the genetic composition of the resident Symbiodinium population can strongly modulate fitness traits of the coral holobiont (Rowan et al., 1997; Sampayo et al., 2008; Voolstra et al., 2009).

Symbiodinium algae provide hosts with photosynthetic products critical for metabolic processes and calcification (Muscatine & Cernichiari, 1969; Muscatine et al., 1984; Trench, 1987). A malfunctioning symbiosis leads to a bleaching event where the algal cells are expelled resulting in a white coloration of the coral (Glynn, 1993; Hoegh-Guldberg, 1999; Hoegh-Guldberg & Smith, 1989). The coral host may or may not recover from a bleaching event (Lang et al., 1992; Marshall & Baird, 2000), which depends on the severity and duration of stress but also can strongly depend on the presence of minor frequency Symbiodinium taxa (i.e., genotypes) in the holobiont prior to bleaching (Berkelmans & van Oppen, 2006; Jones et al., 2008; LaJeunesse et al., 2010a; Putnam et al., 2012). Understanding the flexibility of symbiosis between corals capable of housing a mixed infection (Douglas, 1998; LaJeunesse et al., 2003) versus corals with supposedly strict specificity for one symbiont type (Diekmann et al., 2002; Sampayo et al., 2007) will lead to a better understanding of the ability of corals to survive environmental stressors.

Understanding the factors driving endosymbiont distributions is critical in the assessment of coral reef resilience. Different Symbiodinium species have shown varying photosynthetic efficiency and responses to the amount of light exposure suggesting coral host physiology is at least partially dependent on symbiotic interactions (DeSalvo et al., 2010; Fitt & Warner, 1995; Warner, Fitt & Schmidt, 1996). While the importance of the functional implications of genetic diversity within Symbiodinium clades and subclades and how their presence correlates with environment has been frequently pointed out in literature (Baker, 2003; Knowlton & Rohwer, 2003), only recently is evidence starting to emerge about broad physiological adaptations within Symbiodinium clades (Frade et al., 2008; Van Oppen et al., 2001; Warner et al., 2006). Some general attributes for Symbiodinium genotypes belonging to clades A-C have been proposed. Some members of clades A and B have been more commonly found in high irradiance environments (Rowan et al., 1997; Toller, Rowan & Knowlton, 2001), some clade A lineages have been shown to provide increased UV protection (Reynolds et al., 2008), and members of clade C, the most diverse Symbiodinium lineage are thought to enhance host calcification rates (Cantin et al., 2009; LaJeunesse, 2005). However initial efforts to assign distinct physiological roles of Symbiodinium genotypes have been limited by the resolution of the few available genetic markers and methods. Many of the early clade level generalizations were based on coarse resolution genotyping techniques thereby missing the potential physiological contributions of undetected low frequency Symbiodinium taxa.

It is therefore essential to develop procedures to analyze Symbiodinium at the subclade level in a consistent and quantitative manner. Advances in molecular techniques utilized for Symbiodinium genotyping over the past two decades have resulted in important insights, such as mixed Symbiodinium populations in various coral species and novel symbiont genotypes that previously went undetected (Baird et al., 2007; Baker & Romanski, 2007; Fay & Weber, 2012; LaJeunesse, 2002; Rowan et al., 1997; Thornhill et al., 2014). However, some of the electrophoresis-based methods do not guarantee detection of novel, often low-frequency genotypes, and only confidently detect strains comprising 10% or higher of the total Symbiodinium population (LaJeunesse et al., 2008; Thornhill et al., 2006). Use of next-generation sequencing (NGS) platforms is now gaining popularity as a cost effective, high throughput method capable of discovering and quantifying low frequency strains of Symbiodinium within mixed symbiotic communities, with practical detection limits down to 0.1% (Kenkel et al., 2013; Quigley et al., 2014). Quantitative investigation of low frequency Symbiodinium genotypes, with potential different physiologies, in mixed communities can greatly enhance our understanding of hidden diversity that could potentially play an important role in holobiont fitness (Jones & Berkelmans, 2010; Mieog et al., 2009).

Here we introduce a universally applicable statistical framework to perform such quantitative analysis, and apply it to investigate the Symbiodinium communities within two species of the endangered Caribbean Orbicella annularis species complex (IUCN, 2011), formerly known as a member of the genus Montastraea (Budd et al., 2012). The Orbicella annularis species have been previously shown to host mixed populations of Symbiodinium (Rowan & Knowlton, 1995; Rowan et al., 1997). We used a metabarcoding approach to study two Orbicella species—O. faveolata and O. franksi—at two locations within the Flower Garden Banks National Marine Sanctuary (east and west bank), separated by 19 km of deep water, to see whether the Symbiodinium community variation in these host species is partitioned largely with respect to the host species or with respect to geographic location.

Methods

Locations

The Flower Garden Banks (FGB) is a National Marine Sanctuary established in 1992 and situated 185 km off the coast of Texas (27°54′N, 93°35′W and 27°53′N, 93°49′W for east and west localities, respectively) in the Gulf of Mexico (Fig. 1). The east and west banks are separated by 19 km. Flower Garden Banks are the most northern coral reefs in the Gulf of Mexico making it an important location to understand limits of latitudinal distributions of coral species (Schmahl, Hickerson & Precht, 2008). Twenty-four shallow-water (<50 m) coral species reside at the east and west FGB (Schmahl, Hickerson & Precht, 2008). Compared to other Caribbean reefs, the FGB have lower species diversity, but have much higher coral cover ranging between 50% and 70% (Precht et al., 2005). In addition, the FGB is a relatively deep reef starting at 17 m and extending beyond 45 m (Schmahl, Hickerson & Precht, 2008). Annual average temperatures range between 18 °C and 30 °C providing a unique opportunity to study corals exposed to their thermal minima (Schmahl, Hickerson & Precht, 2008). The remote location of the FGB protects these reefs from most anthropogenic stressors; both land based and recreational.

Figure 1 The study system.

(A) Location of Flower Garden Banks National Marine Sanctuary, Gulf of Mexico (27°54′N, 93°35′W for East Flower Garden Banks and 27°53′N, 93°49′W West Flower Garden Banks) Figure Credit: FGBNMS National Ocean Service (http://flowergarden.noaa.gov/image_library/maps.html). (B) Orbicella faveolata from Panama, Photo Credit: Mónica Medina. (C) Orbicella franksi from Panama, Photo Credit: Mónica Medina.

Coral collections

A total of 197 1 cm × 1 cm coral fragments were collected from the outer edge of Orbicella colonies at both the east and west FGB in August 2011 (O. faveolata, n = 96) and August 2012 (O. franksi, n = 101) with approximately n = 50 per species per bank. Coral tissue was preserved in 96% ethanol and stored at room temperature. Sample depth ranged from 21 to 23 m.

Laboratory procedures and host genotyping

FGB holobiont DNA was isolated following the phenol-chloroform protocol described in Davies et al. (2013). One hundred ninety-three coral hosts were successfully amplified at nine microsatellite loci (Davies et al., 2013). STRUCTURE (version 2.3.4) output (q-score) (Falush, Stephens & Pritchard, 2003; Falush, Stephens & Pritchard, 2007; Hubisz et al., 2009; Pritchard, Stephens & Donnelly, 2000) was used to identify non-hybrid coral colonies. Hybrids from the O. annularis species complex have been reported in literature (Budd & Pandolfi, 2004; Fukami et al., 2004; Szmant et al., 1997). Only individuals with greater than 80% posterior probability of belonging to one of the two major STRUCTURE derived clusters were retained (73 samples of O. faveolata and 101 samples of O. franksi) (Foster et al., 2012). Sixty of these, fifteen colonies of O. faveolata and fifteen colonies of O. franksi from both east and west FGB, were chosen for Symbiodinium ITS-2 genotyping. To look for genetic structure among coral populations between the two locations (east and west banks), an admixture model was run starting with a uniform alpha for degree of admixture, uncorrelated allele frequencies for five simulations, a burn-in of 300,000 steps and 106 Markov-Chain Monte Carlo (MCMC) iterations. STRUCTURE results were then used as input to run STRUCTURE HARVESTER to select the optimal number of clusters (K) (Earl & vonHoldt, 2012; Evanno, Regnaut & Goudet, 2005). Using CLUMPP (Jakobsson & Rosenberg, 2007), output files from STRUCTURE HARVESTER were used to combine the results of replicated runs by computing weighted averages followed by plotting the results using DISTRUCT (Rosenberg, 2004). To assess within species differentiation, each species was analyzed separately in STRUCTURE applying the same parameters for all analyses (Foster et al., 2012). An analysis of molecular variance (AMOVA) was implemented in GenAlEx (version 6.5) to assess genetic differentiation by computing pairwise FST for species and sites (Peakall & Smouse, 2012).

Amplification of ITS-2 for 454 sequencing

ITS-2 was amplified in each of the sixty individual hosts and submitted for deep amplicon sequencing in January 2013 using Symbiodinium specific ITS-2 primers, ITS-Dino-forward (5′-GTGAATTGCAGAACTCCGTG-3′) (Pochon et al., 2001) and its2rev2-reverse (5′-CCTCCGCTTACTTATATGCTT-3′) (Stat et al., 2009). The target amplicon was approximately 300 base pairs long. Each 30 µL PCR reaction contained 13.3 µL of water, 3.0 µL 10 × ExTaq HS buffer, 0.2 mM dNTP, 0.75 U ExTaq HS polymerase (Takara Biotechnology), 0.375 U Pfu polymerase (Agilent Technologies), 0.2 µM final primer concentration and 50 ng of DNA template (Kenkel et al., 2013). A DNA Engine Tetrad 2 Thermal Cycler (Bio-Rad, Hercules, CA, USA) was used for all amplifications. The following PCR protocol was used: 20 cycles of 94 °C for five minutes, 95 °C for 40 s, 59 °C for two minutes, 72 °C for one minute and final extensions of 72 °C for five minutes. PCR product intensity of all individuals was determined on one two percent gel-red stained agarose gel. Additional cycles were added to individuals to obtain the same uniform band intensity (just above the visibility threshold on an agarose gel) and the final cycle number was recorded. Individuals that had not amplified by 35 cycles were repeated using a lower starting template (20 ng/µL) to reduce the inhibition by contaminants. All individuals amplified by 34 cycles except one west FGB O. faveolata and one east FGB O. faveolata which were removed from the analysis. PCR products were cleaned using GeneJET PCR purification kits (Fermentas Life Sciences). Six individuals were randomly selected and run on a two percent agarose gel to ensure sufficient DNA quantities remained post clean-up.

New 30 µL PCR reactions were performed to incorporate A and B Rapid adaptors specific for 454 GS FLX. The adaptors designs were: the reverse barcoded primer sequence (A-Rapid primer + unique barcode + its2rev2 primer) and forward B-rapid primer (B-Rapid primer + ITS-Dino) (Fig. S1). A unique barcode was assigned to each amplified coral host individual (n = 58). Each reaction contained 50 ng of cleaned PCR product, 17.6 µL water, 0.2 mM dNTP, 3 µL 10 × ExTaq HS buffer, 0.75 U ExTaq HS polymerase (Takara Biotechnology), 0.375 U Pfu polymerase (Agilent Technologies), 50 ng of PCR product, 0.33 µM of 454 B-Rapid ITS2-forward (5′-CCTATCCCCTGTGTGCCTTGAGAGACGHC+ GTGAATTGCAGAACTCCGTG-3′) and 0.33 µM of 454 A-Rapid ITS2 adaptor with unique barcode (5′-CCATCTCATCCCTGCGTGTCTCCGACGACT+ TGTAGCGC + CCTCCGCTTACTTATATGCTT-3′, example barcode sequence is shown in bold) (Kenkel et al., 2013). PCR was performed on a DNA Engine Tetrad 2 Thermal Cycler (Bio-Rad, Hercules, CA, USA) under the following conditions: 95 °C for five minutes, four cycles of 95 °C for 30 s, 59 °C for 30 s, 72 °C for one minute followed by incubation at 72 °C for five minutes. Samples were verified on one two percent agarose gel and pooled based on band intensity. Pools were ethanol precipitated. Three to five micrograms of the cleaned product was run on a one percent SYBR Green (Invitrogen) stained gel. The target band was excised using a blue-light box and soaked in 25 µL of milli-Q water overnight at 4 °C. The supernatant was submitted and sequenced at the University of Texas-Austin Genome Sequencing and Analysis Facility (GSAF) aiming to obtain two thousand reads per sample.

Bioinformatic pipeline to obtain read count data

We provide a detailed bioinformatics walkthrough (Data S1). This and the updated versions of the pipeline will be hosted on Matz Lab Methods web page (http://www.bio.utxas.edu/research/matz_lab/matzlab/Methods.html). Briefly, the pipeline relies on three widely used software packages—fastx_toolkit (http://hannonlab.cshl.edu/fastx_toolkit/), usearch (Edgar, 2010) and SHRiMP2 (David et al., 2011) along with additional custom Perl scripts to streamline processing of multiple read files. The pipeline can be run on any Linux, UNIX or Mac OS-X computer. It includes four steps: (i) quality and length filtering of reads, (ii) identifying haplotypes (or operational taxonomic units, OTUs), (iii) mapping the reads to the newly assembled haplotype database, and (iv) deriving counts of unambiguous matches to each haplotype within each sample. The pipeline is suitable for fasta or fastq formatted read files, providing maximum versatility for use with the currently dominant NGS technologies. The OTUs were used as queries for BLASTn (Altschul et al., 1990) and were aligned between each other using Clustal Omega online server (version 1.2.0) (Goujon et al., 2010; McWilliam et al., 2013; Sievers et al., 2011). Alignments were examined and manually edited using SeaView (version 4.4.2) (Gouy, Guindon & Gascuel, 2010). The term ‘genotype’ will hereafter be used to reference previously published Symbiodinium clade types where the term ‘haplotype’ will reference the Symbiodinium sequences represented in our collected dataset.

Model-based analysis: theory and implementation

The OTU counts analysis follows the methodology described earlier (Matz, Wright & Scott, 2013), implementing a Poisson-lognormal generalized linear mixed model to jointly infer parameters of interest such as OTU-specific abundance changes in response to fixed factors as well as nuisance parameters such as variation attributable to the random differences in sequencing depth among samples. Incorporation of the Poisson component into the model makes it possible to adequately analyze sparsely represented haplotypes that might be missing from many samples due to undersampling rather than true absence.

Let yoijk be the count for OTU o under condition i, group j, sample k. The model assumes that this count arises from a Poisson-lognormal distribution: (1) yoijk∼PLNλoijk,σ02

where PLN(m, v) denotes the Poisson-lognormal distribution with rate parameter m and log-variance v. In our model, the log-variance is OTU-specific, and the rate term involves regressions on both fixed and random effects.

The model for log-rate term ψoijk = log(λoijk) takes the form (2) ψoijk=Sijk+Io+Boi+ck+aoj;Sijk=Is+Bis+ck+ajs

where Io represents an OTU-specific intercept, Boi is the fixed effect of condition i on OTU o, (examples of condition being host species, location, or treatment), ck is a random effect meant to capture the deviation in sequencing coverage in sample k, and aoj is the OTU-specific random effect associated with the jth level of some grouping variable (tank, block, or plot) and/or individual if the design involved repeated measures. The ck and aoj are random effects modeled as samples from Gaussian distributions; in case of ck it is the same distribution for all OTUs and in case of aoj the distributions have OTU-specific variances. Both Boi and aoj terms represent design matrices that can potentially include multiple factors as well as their interactions, or none at all.

The model in (2) is different from the one described in Matz, Wright & Scott (2013) in two minor aspects. First, it does not include the secondary error term, the OTU-specific effect of sample, which would be hard to estimate without technical replicates (rare in metabarcoding); the variance that would be attributable to this term is absorbed within the haplotype-specific residual error term σo2 one level up in the model hierarchy (Eq. (1)). Another difference is the term Sijk that describes the behavior of an artificial OTU with counts equal to the sum of all counts in a sample. This “sum-OTU” is modeled together with all other OTUs, decomposing its variance into its own intercept, fixed and random components (bottom row in (2)). Its special role in the model (2) is that, after the model is fitted, the fixed OTU-specific effects (Bih) are being expressed relative to the ones for the sum-OTU, which controls for possible systematic biases in coverage across experimental conditions. It also makes it possible to express the inferred OTU abundances as proportions of total, which is natural in metabarcoding.

The model is fitted to the data using a Bayesian Markov chain Monte Carlo (MCMC) procedure utilizing non-informative over-dispersed priors for fixed effects, non-informative inverse-Wishart priors (variance V = 1, degree of belief nu = 0) for variance components ck and aOJ and the weakly informative inverse-Wishart prior (V = 1 and nu equal number of OTUs minus 0.998) for the residual haplotype-specific error σo2 (Hadfield, 2010). This formulation results in well-mixed MCMC chains and the ability to handle up to about 200 OTUs on a personal computer. The OTUs that are so sparsely represented that no reliable parameter estimates could be obtained for them are detected based on high autocorrelation among sampled values and are discarded from further analysis.

The statistical approach described above has been implemented in a user-friendly R package (MCMC.OTU) (R Developmental Core Team, 2013). The official releases of the package are available from the Comprehensive R Archive Network (CRAN, http://cran.r-project.org/), while the beta-versions and the walkthrough script based on the data from this paper are available from Matz Lab Methods web page (http://www.bio.utexas.edu/research/matz_lab/matzlab/Methods.html).

In addition to the model-fitting function, the package MCMC.OTU includes functions for data reformatting, selecting quantifiable OTUs based on proportion of global count, identifying OTUs for which the model generated reliable parameter estimates and for converting raw counts into log-transformed normalized data for principal component analysis and studies of among-OTU correlations (this transformation is described in the next section). MCMC.OTU package also includes functions to calculate statistical significance for all possible pairwise comparisons and to plot results. The package includes extensive documentation and examples and is designed to make the analysis accessible to the novice R user.

Statistical analysis: Orbicella data

The R script (with detailed comments) that was used to analyze Orbicella data is provided (Data S2). Briefly, the analysis involved four steps. First, outlier samples and OTUs are discarded. The criterion for discarding samples was the total log-counts being less by ≥2.5 standard deviations than the mean across all samples; this resulted in discarding of a single sample of O. faveolata from the east bank. The OTUs were deemed quantifiable if they comprised at least 0.1% of global sum of counts, as per Quigley et al. (2014). Second, the data are reformatted and the model is fitted; in this case the model included fixed effects of host species (O. faveolata or O. franksi), bank (east or west) and their interaction. Third, pairwise differences between all combinations of fixed effects are calculated based on their sampled posterior distributions, their statistical significances are calculated as described in Matz, Wright & Scott (2013) and adjusted for multiple testing using false discovery rate (FDR) correction (Benjamini & Hochberg, 1995). Fourth, the results are visualized as trellis plots.

For principal components analysis, all OTUs detected in greater than 10% of all samples were retained. The counts data were normalized by multiplying them by the sample size factor, which was computed as the ratio of the sum of counts in the sample to the mean sum of counts across samples, and log10-transformed using a “started log” method, in which the zero counts are replaced prior to log-transforming by a certain positive value (in this case 0.1). The MCMC.OTU package also implements the “log-linear hybrid” transformation (Rocke & Durbin, 2003), which in downstream analyses gives very similar results to the ones obtained with the “started log” transform. The transformed data were subjected to principal components analysis using R (package ‘vegan’ Oksanen et al., 2013). The log-transformed normalized data in which the original zero count datapoints were left undefined were used to analyze pairwise correlations between haplotype abundance across samples with the assumption that OTUs representing the same genome (paralogs) are likely to show positive correlation, whereas negative correlation between two OTUs indicates that they represent different genomes (Kenkel et al., 2013).

Analysis of structural conservation

Our identified seven B1 haplotypes were aligned with the 19 identified B1 ITS-2 sequences from the ITS-2 database (http://its2.bioapps.biozentrum.uni-wuerzburg.de/) and trimmed to be of equal length. The pooled dataset of the identified 19 B1 ITS-2 sequences was then used to predict the structure for the trimmed seven B1 haplotypes using the ‘model’ tool (Koetschan et al., 2010; Koetschan et al., in press; Schultz et al., 2006; Selig et al., 2008; Wolf et al., 2005). The seven OTUs were aligned with the B1 model sequences used to predict secondary structure using the ‘sequence and structure’ tool. The resulting aligned fasta file of sequences with structure (Data S3) was opened in 4SALE to compare secondary structures looking for the four conserved attributes of the ITS-2 gene secondary structure (Schultz et al., 2005) and compensatory base changes (Seibel et al., 2006; Seibel et al., 2008).

Results

Host genetics

STRUCTURE analysis detected genetic differences between the two coral species, but no divergence between locations for either of them (Fig. 2). Output files from STRUCTURE HARVESTER showed a delta K of two for all analyses except the independent analysis of Orbicella faveolata (n = 73) which showed a delta K of three (Fig. S2). This result was verified by AMOVA analysis (Table 1). AMOVA results comparing FST between sites showed no significant genetic differentiation in Orbicella franksi (Table 1). Orbicella faveolata showed small, but significant FST (p < 0.02) between banks (Table 1), which was also reflected on the STRUCTURE plot (Fig. 2). There was a much clearer genetic differentiation between species (Fig. 2) with a significant FST of 0.07 (Table 1).

Figure 2 Genetic analysis of the coral host.

DISTRUCT plots from STRUCTURE, K = 2 except where noted. (A) All Orbicella faveolata and Orbicella franksi samples collected from East and West Flower Garden Banks National Marine Sanctuary, Gulf of Mexico (n = 193). (B) Same as ‘A’ but potential hybrids removed (n = 174). (C) Only Orbicella faveolata but with potential hybrids removed (n = 73, K = 3). (D) Only Orbicella franksi but with potential hybrids removed (n = 101). (E) The selected 60 Orbicella faveolata (n = 30) and Orbicella franksi (n = 30).

Table 1 Analysis of Molecular Variance (AMOVA) Fixation index (FST) values.

Analysis of Molecular Variance (AMOVA) Fixation index (FST) values showing genetic diversity results among Orbicella faveolata and Orbicella franksi, among Orbicella franksi within the two geographic locations and among Orbicella faveolata within the two geographic locations.

	FST	p-value	
O. faveolata vs O. franksi	0.069	0.001	
O. franksi East vs West	0	0.529	
O. faveolata East vs West	0.009	0.016	

Symbiodinium metabarcoding summary

ITS-2 sequencing yielded 170,344 raw reads for 58 individuals, averaging 2,937 reads per individual (Table 2). After removing all reads shorter than 290bp, 122,867 reads representing 20,260 unique sequences remained. The clustering was performed at the 97% similarity level and resulted in 153 OTUs. The relatively high clustering threshold was chosen to maximize the power to detect low-divergence haplotypes, since in Symbiodinium even a few bases of difference in ITS-2 sequence could delineate ecologically distinct strains (e.g., LaJeunesse et al., 2004; LaJeunesse et al., 2003). All filtered reads were unambiguously mapped against this OTU collection. The sum of unambiguous match counts per sample ranged from 256 to 5,394, with median at 1,884 and 25% and 75% quartiles at 1,316 and 2,597, respectively. The sample with 256 counts was subsequently discarded, leaving 864 as the next lowest sum of counts per sample.

Table 2 The sequencing coverage and mapping efficiency by geographic location and species.

Summary of ITS-2 amplicon sequencing for two coral host species collected from Flower Garden Banks, Gulf of Mexico using the 454 GS FLX platform. Individuals are summed by geographic location and species.

	Raw read
number	Trimmed
reads	Mapped
reads	Mapping
efficiency	
East	95,478	68,670	68,637	100%	
West	74,871	54,197	54,175	100%	
O. faveolata	74,840	53,938	53,913	100%	
O. franksi	95,509	68,929	68,899	100%	
Total	170,349	122,867	122,812	100%	

Model-based analysis of haplotype abundances

Of the 153 OTUs identified, ten were deemed amenable to quantitative analysis since each of them exceeded 0.1% of the total counts (Quigley et al., 2014), but for only seven of them reliable parameter estimates could be obtained since the other three were too sparse among samples (found in eight or less samples out of 57). These seven OTUs, hereafter referred to as haplotypes, were analyzed further. Haplotype II was by far the most dominant accounting for 90.06% of all reads mapped across all individuals (Fig. 3). Poisson-lognormal generalized linear mixed model analysis revealed that haplotype IV was significantly (FDR < 0.026) diminished at the west bank in both species, haplotype VI was diminished in O. faveolata at the east bank (FDR = 0.001), and haplotype VII was elevated in the west bank in both species (FDR < 0.022) (Fig. 4 and Table S1).

Figure 3 The percentage of reads for the seven minor Symbiodinium haplotypes by geographic location and species.

The dominant Symbiodinium B1 haplotype II comprised the remaining bulk of reads. Total read numbers were: East: Orbicella faveolata = 26,916 reads, East: Orbicella franksi = 40,094 reads, West: Orbicella faveolata = 26,264 reads, West: Orbicella franksi = 27,503 reads.

Sequence similarities of the detected haplotypes

All seven haplotypes best matched Symbiodinium clade B type B1 (JN 558059.1) (Pochon et al., 2012), recently identified as Symbiodinium minutum (AF 333511.1) (LaJeunesse, Parkinson & Reimer, 2012). After end-trimming to equalize length, haplotypes I and II matched B1 (JN 558059.1, AF 333511.1) (LaJeunesse, Parkinson & Reimer, 2012; Pochon et al., 2012) with 100% identity, whereas the remaining five haplotypes did not find an exact match in the database (Data S1 and Fig. S3, Gouy, Guindon & Gascuel, 2010). Haplotype III differs from B1 by a 13 base pair deletion. Haplotype IV differs from B1 (JN 558059.1, AF 333511.1) (LaJeunesse, Parkinson & Reimer, 2012; Pochon et al., 2012) by a ten base pair insertion. Haplotype V differs from B1 (JN 558059.1, AF 333511.1) (LaJeunesse, Parkinson & Reimer, 2012; Pochon et al., 2012) by nine different single nucleotide insertions. Haplotype VI differs from B1 (JN 558059.1, AF 333511.1) (LaJeunesse, Parkinson & Reimer, 2012; Pochon et al., 2012) by a nine base pair deletion and thymine insertion at position 156. Haplotype VII differs from B1 (JN 558059.1, AF 333511.1) (LaJeunesse, Parkinson & Reimer, 2012; Pochon et al., 2012) by an eleven base pair deletion. These indels do not occur in homopolymer regions and are unlikely to be the results of 454 sequencing error (Margulies et al., 2005).

Figure 4 Distribution of abundance of the three significantly differentially represented Symbiodinium haplotypes by geographic location (A) and coral species (B).

We additionally used NCBI nBLAST to identify the top database match for all 153 OTUs. The quantitative analysis of these OTUs was not possible due to their extremely sparse representation among our samples. Moreover, some of these rare OTUs were obvious contaminants. We detected 130 top hits with B1s, one B19a, three B3, one B2, one C90, one C91, one D1, one D2, one F5.2b, two G6, one G, one host contamination (Orbicella franksi) and several ITS-2 sequences for free-living dinoflagellates species.

Potential paralogous copies of novel Symbiodinium haplotypes detected

Correlations between haplotype abundances across samples indicates whether the haplotypes might be paralogous loci from the same genome (positive correlation) or not (negative correlation) (Kenkel et al., 2013). Of the seven quantifiable haplotypes only III and V showed a significant positive correlation (p = 0.016). With the exception of these two haplotypes, correlation analysis did not find evidence that any other of the haplotypes might be paralogous. Instead, it confirmed genomic independence for some of them via negative correlations: haplotypes IV and VI showed significant negative correlation with the major haplotype II (p < 0.001 and 0.023, respectively), and haplotype VII was negatively correlated with haplotype I (p = 0.0021) and VI (p = 0.032) (Fig. S4).

Pseudogene analysis

Three B1 model sequences with structure from the ITS-2 database were used to predict the secondary structure for the seven haplotypes and top blast hit Symbiodinium B1 (JN 558059.1). Only 72.7% of the third helix for the ITS-2 reference (GI 156466871) was found in the predicted secondary structure of haplotype III, while the fourth helix of haplotype V was only 85.7% complete compared to the reference structure (GI 156466870) (Table S2). The remaining haplotypes all had 100% conservation in all four helices with the model B1 sequence with structure (GI 58081212). Using 4SALE (Seibel et al., 2006; Seibel et al., 2008) no compensatory base changes (CBC) were detected when comparing haplotypes to their reference B1 sequence from the ITS-2 database. Visual comparisons using 4SALE structure viewer for haplotype III and V with their model B1 sequence showed high conservation across sites. However, given the reduced homology modeling results for haplotype III and V coupled with their positive correlation among samples, we predict that at least one of them (most likely haplotype III) is a paralog potentially representing a non-functional copy of the rRNA gene.

Model-free approach

The first component (PC1) from the PCA explained 18.64% of the variation and principle component two (PC2) explained 11.82% of the variation. Retaining the first six components meets Kaiser’s criterion (Kaiser, 1960), defined as all components with a standard deviation greater than one, and explain 62.67% of the variation. The samples were visibly partitioned with respect to the sampling locality with loadings corresponding to the three model-inferred significant haplotypes underscoring the partitioning (Fig. 5).

Figure 5 Principle Components Analysis (PCA) of Symbiodinium read data for all haplotypes accounting for >0.1% of total.

First two principal components explain 30.46% of the variation (n = 57). Arrows are loadings for the three haplotypes significantly differentially represented between East and West banks according to the GLMM model.

Discussion

Host species delineation

Nine recently developed microsatellite markers (Davies et al., 2013) were used for host genotyping to distinguish the two host species, Orbicella faveolata and O. franksi, since these species have been shown to hybridize (Budd & Pandolfi, 2004; Fukami et al., 2004; Szmant et al., 1997). Species in the Orbicella annularis complex vary considerably morphologically, however genetic differences are not as pronounced making species identification in the Orbicella annularis species complex challenging (Fukami et al., 2004). Microsatellite data were analyzed using STRUCTURE (Fig. 2) and AMOVA (Table 1) to ensure that the selected individuals were not recent hybrids.

Detecting novel symbiont haplotypes

Quigley et al. (2014) verified the detection sensitivity of the method used here for genotypes from clades A, C and D down to 0.1%. In our study, five of the seven quantifiable haplotypes did not have an exact match in the NCBI sequence database and therefore were potentially novel. It remains to be seen whether these haplotypes are endemic to the FGB, since this metabarcoding approach has not yet been widely applied across the Caribbean. Another alternative (and a typical concern in metabarcoding) is that the detected rare haplotypes represent artifacts of amplification, sequencing or bioinformatics analysis, which is particularly probable in cases such as ours when OTUs are expected to be of low sequence diversity, necessitating the use of high sequence identity thresholds (97%) during OTU inference. To deal with this issue to some degree, we chose a conservative approach while selecting haplotypes for further analysis. First, guided by the Quigley et al. (2014) results we required the candidate haplotypes to be present in appreciable counts (>0.1% of total, which amounts to about one hundred for our dataset). Second, our methodology implies that the haplotypes must be found across multiple samples to be amenable for quantification. These two criteria are expected to considerably lower the chance for the inferred haplotypes to be artifacts of sequencing or PCR. An additional assurance is provided by the fact that each of the minor haplotypes was discriminated by multiple small indels or a single large (≥10 bp) indel (Fig. S3), which are unlikely to happen as a result from sequencing or PCR errors. Further improvement of the reliability of detected OTUs can be obtained by sequencing independently prepared technical replicates, which we intend to explore in the future.

Another concern when using a multi-copy marker undergoing concerted evolution such as ITS-2 (Koch, Dobeš & Mitchell-Olds, 2003; Stat et al., 2011) is that some of the detected haplotypes could be paralogs or pseudogenes from the same genome (Thornhill, Lajeunesse & Santos, 2007) rather than true representative of unique symbiont strains. To address this issue, our method follows Kenkel et al. (2013) in exploring pairwise correlations among haplotype abundances. Haplotypes potentially from the same genome are expected to show positive correlations across samples, whereas a negative correlation indicates that the haplotypes are from different, unique genomes. In our dataset, only two of the seven haplotypes (III and V) show significantly positive correlation and therefore are likely to be paralogs from the same genome. The remaining haplotypes are either uncorrelated or show negative correlations (Fig. S4). Only until single-copy genetic markers specific for Symbiodinium are developed will we be able to fully overcome this problem (LaJeunesse & Thornhill, 2011).

Quasi-monotypic symbiont population in Orbicella species at the FGB

Even though multiple haplotypes have been detected in both O. faveolata and O. franksi, they were all closely related to clade B type B1, the most prevalent Symbiodinium type in the Caribbean (Baker, 2003; LaJeunesse, 2002; LaJeunesse et al., 2003). This result is unusual for the genus Orbicella, since other assessments of Symbiodinium diversity in Orbicella spp. throughout the Caribbean have shown mixed populations of Symbiodinium clades ranging from A to D (Rowan & Knowlton, 1995; Rowan et al., 1997; Thornhill et al., 2006; Toller, Rowan & Knowlton, 2001). A variety of environmental factors have been proposed to explain Symbiodinium distributions, including but not limited to depth (LaJeunesse, 2002), irradiance levels (Fitt & Warner, 1995), latitudinal location (LaJeunesse et al., 2003) and temperature (LaJeunesse et al., 2010b). The fact that Orbicella species from the FGB harbor reduced symbiont diversity could be explained by the fact that FGB is a marginal coral reef environment. The FGB represent the northernmost latitudinal reef in the Gulf of Mexico (LaJeunesse & Trench, 2000) and experiences lower thermal minima relative to the rest of the Caribbean (Schmahl, Hickerson & Precht, 2008; Thornhill et al., 2008). In addition, FGB is one of the deepest reefs at which Orbicella spp. are found (>17 m), and it has been previously shown that corals from deeper environments have fewer mixed infections (LaJeunesse, 2002). Whether the within-clade haplotype diversity detected in this study has physiological implications for host fitness remains unknown. This question could be addressed with future association studies and manipulative experiments using faster evolving loci, such as microsatellites, for a more complete characterization of the fine scale genetic diversity of Symbiodinium (Finney et al., 2010; Pettay & LaJeunesse, 2007; Santos et al., 2004; Thornhill et al., 2014).

Symbiodinium variation between two geographic locations

Three minor Symbiodinium haplotypes (IV, VI and VII) represented at the level of 0.1%–10% of the total were significantly different in abundance between sampled locations, but not between host species (Fig. 4). Previous studies of broadcast-spawning corals have shown genetic differentiation of Symbiodinium across relatively small geographic distances (Howells et al., 2012; Howells, Oppen & Willis, 2009; Kirk et al., 2009). Another study also showed strong genetic divergence in Symbiodinium communities and in host species across different habitat types (Bongaerts et al., 2010). Still, our results are unexpected given significant genetic differentiation between the host species (Fig. 2 and Table 1) and the fact that the sampled locations are in close proximity and are characterized by similar environmental conditions (Schmahl, Hickerson & Precht, 2008). The detected difference in Symbiodinium between east and west banks could be due to random drift in isolation: 19 km of deep water with no “stepping stone” habitats could present a barrier to Symbiodinium dispersal, which remains a poorly understood process. Alternatively, the differences could be due to subtle variation in environmental conditions. However, this latter explanation requires no genetic recombination between the detected Symbiodinium haplotypes. Only then would the environmental selection be likely to alter the relative frequencies of whole genomes of these detected haplotypes including the ITS-2 marker. These alternatives will be interesting to investigate in the future using higher-resolution genetic markers for Symbiodinium (e.g., microsatellites), especially in conjunction with holobiont physiology.

Detecting Symbiodinium community shifts

Two mechanisms have been proposed to explain the plasticity of symbiosis between host and symbiont termed “shuffling” and “switching”. “Shuffling” is a change in the existing proportions of a mixed Symbiodinium infection whereby a dominant symbiont type may become reduced while a background symbiont type becomes prevalent (Berkelmans & van Oppen, 2006; Fay & Weber, 2012; LaJeunesse et al., 2009; Rowan et al., 1997; Silverstein, Correa & Baker, 2012; Stat, Carter & Hoegh-Guldberg, 2006). “Switching” is a postulated event reflecting the idea of an “open” symbiotic system when a new exogenous Symbiodinium becomes the dominant type (Baker, 2001; Buddemeier & Fautin, 1993). In order to assess whether corals “switch” or “shuffle”, we must consistently and confidently detect the cryptic Symbiodinium diversity. Use of a highly sensitive and quantitative genotyping method will allow for the assessment of distribution patterns of Symbiodinium-host relationships across spatial scales. By doing so, it will also become more feasible to examine changes in Symbiodinium composition over time and detect species shuffling as well as potential horizontal uptake with finer spatio-temporal resolution. The quantification methodology described here and in recent publications (Kenkel et al., 2013; Quigley et al., 2014) will contribute to a better understanding of the plasticity of symbiosis and community composition shifts. This ITS-based Symbiodinium metabarcoding protocol utilizes primers common to all Symbiodinium clades and therefore permits not only detection of known genotypes, but also discovery of new ones. Recent studies are also starting to apply this approach to investigate other members of the coral holobiont (Rohwer et al., 2002), such as other algae (Davies et al., 2014), fungi, protists (Cumbo et al., 2013), bacteria (Ceh, Van Keulen & Bourne, 2010; Morrow et al., 2012; Sunagawa et al., 2009), archaea (Beman et al., 2007) and viruses.

MCMC.OTU package: model-based approach to quantitative metabarcoding

This paper provides a universal model-based statistical framework for analyzing the quantitative shifts in community composition based on counts data. The main advantage of our approach over previously developed techniques (Paulson et al., 2013; Segata et al., 2011; White, Nagarajan & Pop, 2009) is the ability to directly analyze haplotype counts (rather than haplotype proportions) using an over-dispersed Poisson-lognormal model. This properly accounts for the possibility of a sparsely represented haplotype being absent due to under-sampling rather than due to any of the modeled effects. Furthermore, the method does not require data normalization since the model accounts for the effects of unequal sequence coverage (term ck in Eq. (2)). The remaining residual biases are corrected by expressing the inferred OTU-specific effects relative to those for the artificial “sum-count” OTU (term Sijk in Eq. (2)). The method draws power from analyzing all OTUs within a single model, and is very versatile in terms of experimental design, allowing for any combination of fixed and random effects. This methodology has been derived from recent advancements in analysis of quantitative PCR data (Matz, Wright & Scott, 2013) by incorporating modifications specific for metabarcoding such as the use of “sum-count” OTU and the assumption that the data are unlikely to include technical replicates (although the method can properly handle these as well). The method is not restricted to ITS or Symbiodinium and can feasibly analyze up to about 200 OTUs on a personal computer. We hope that it will be useful for a wide range of metabarcoding studies.

Conclusions

We present a novel methodology for quantitative analysis of metabarcoding data and apply it to assess Symbiodinium diversity at the remote Flower Garden Banks. Results show coral hosts Orbicella faveolata and O. franksi harbor Symbiodinium type B1, represented by seven haplotypes. Two of these haplotypes were significantly diminished at the west FGB and one was significantly diminished at the east FGB. Future work using faster evolving loci, such as Symbiodinium-specific microsatellites, may show variations between host species or geographic locations within clade B lineages. Wider use of metabarcoding, not only with ITS-2 but with additional loci, to quantify Symbiodinium genetic diversity within multiple hosts will significantly advance our understanding of the complex dynamics of coral-Symbiodinium symbioses.

Supplemental Information

Figure S1 Rapid-barcode primer design

Click here for additional data file.

Data S1 A bioinformatics walk through document, custom perl scripts and the Clustal Omega alignment file for the seven Symbiodinium haplotypes and Symbiodinium B1 (JN558059.1)

Click here for additional data file.

Data S2 The MCMC.OTU package (version 1.0.7) with a walk through script to analyze the dataset discussed here

Click here for additional data file.

Data S3 A fasta file of the aligned seven Symbiodinium haplotypes, Symbiodinium B1 (JN558059.1) and the three B1 template sequences from ITS-2 and their predicted structure

Click here for additional data file.

Figure S2 Delta K plots from STRUCTURE HARVESTER

Click here for additional data file.

Table S1 Table of log10-fold changes and FDR-corrected p-values inferred by the GLMM model

Click here for additional data file.

Figure S3 Alignment of seven detected haplotypes and top BLAST hit Symbiodinium B1 (JN 558059.1)

Click here for additional data file.

Table S2 The percent transfer structure results from using ITS-2 ‘model’ tool to predict the secondary structure of our identified B1 ITS-2 sequences

Percent transfer is the percent of each helix from the model B1 ITS-2 structure that was found within the secondary structure for a novel ITS-2 sequence.

Click here for additional data file.

Figure S4 Pairwise correlations among the seven identified haplotypes

Scatter plots with lowess-smoothing lines are shown in lower triangle and p-values < 0.1 are shown in the upper triangle.

Click here for additional data file.

We are grateful to personnel at the FGBNMS (E Hickerson & GP Schmahl) for permits (FGBNMS-2009-005-A2, A3), boat time and help during fieldwork. We also thank Eli Meyer for help with Orbicella sample collections. We also appreciate Michele Weber and Anke Kleuter for assistance editing and their expertise in Symbiodinium genetic diversity, Bishoy Kamel for bioinformatics support and Dr. Scott Hunicke-Smith and staff at the Genomics Sequencing and Analysis Facility at University of Texas at Austin for technical support.

Additional Information and Declarations

Competing Interests

Author Contributions

Field Study Permissions

DNA Deposition

The authors declare there are no competing interests. Dr. Monica Medina is an Academic Editor for PeerJ.

Elizabeth A. Green conceived and designed the experiments, performed the experiments, analyzed the data, wrote the paper, prepared figures and/or tables, reviewed drafts of the paper.

Sarah W. Davies conceived and designed the experiments, performed the experiments, wrote the paper, reviewed drafts of the paper.

Mikhail V. Matz conceived and designed the experiments, analyzed the data, contributed reagents/materials/analysis tools, wrote the paper, reviewed drafts of the paper.

Mónica Medina conceived and designed the experiments, contributed reagents/materials/analysis tools, wrote the paper, reviewed drafts of the paper.

The following information was supplied relating to ethical approvals (i.e., approving body and any reference numbers):

Flower Garden Banks National Marine Sanctuary permit FGBNMS-2009-005-A2, A3.

The following information was supplied regarding the deposition of DNA sequences:

Raw sff files were uploaded to Sequence Read Archive (SRA) Accession Number 245412.

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
