# Peer review of "Quantifying cryptic Symbiodinium diversity within Orbicella faveolata and Orbicella franksi at the Flower Garden Banks, Gulf of Mexico"

_PeerJ, doi:10.7717/peerj.386_

## Round 0.1 · original submission · Major Revisions

Dear Authors, Prof. Medina; please correct your paper according to all the referees comments and resubmit the revised version of your paper.

Reviewer 1 ·

Basic reporting

The overall reporting of the article is fine and it is of easy comprehension. It makes a good introduction and discussion on the theme subject. There are some minor comments on the lines below:

- The statistical analysis section (lines 172-183) is over-oriented by the R libraries and packages used. It would make an easier reading if the sentences were guided by the analysis itself and the libraries cited in parenthesis.
- The major loadings to each PC1 and PC2 on PC Analysis could be reported (lines 214-218). Those (probably abundances of haplotypes IV and V) could match the GLMM results.

“Quigley, KM 'unpublished data'” should not be referenced so many times, as most times the reference for Kenkel et al (2013) is enough. There is a valid reference for Quigley on the discussion about methodological sensitivity (Line 313). But even in this section, a theoretical detection limit for relative abundance could be estimated for the article data, based on the mean sequencing throughput and trimming criteria.

Line 21. change "Dependent" to "Depending"
Line 36. It's not clear what "saturation points" stands for (probably light?)
Line 114. There's a "," before "each"
Lines 138-140. This last sentence should not be in the methods, at most in the discussion. Anyway I would take it out of the article as it is not a big problem for the analysis of overall diversity presented (it would be a problem if the purpose was to compare relative diversity between samples).
Line 192-193. It is not clear if the sentence is referring to differences between species (False) or between sites for each species (True).
Lines 319-321. I would not use Illumina as an example on the discussion if the 454 technology was used for the article. Illumina citation can be replaced by an indefinite citation to NGS and multiplexing.
Line 326. The section header “Limitations of deep amplicon sequencing” should be removed or altered. The limitations on the paragraph starting on line 327 relates to ITS2 being a multi-copy marker. The discussion in important but this problem will arise in any methodology for delineating species based on ITS2 sequences.
Line 601. Reference title is written in capital letters

Experimental design

No comments

Validity of the findings

The main critic I have to the article is the selection criteria on line 167:
“Of 153 OTUs identified, only five OTUs had a median count exceeding one (i.e., were detected in more than half of all samples) and were retained.”

By this criteria, Symbiodinium species that were dominant in a few colonies but absent on more than half of the samples must have been excluded from the analysis. This criteria will only allow you to recover widespread sequences and shall prevent you to reach the main objective of the article, to reveal the cryptic diversity announced.

On Figure 3 it is said that haplotype II was dominant in 93.26% of the samples, but what happens to the other 6.74%? Were they colonized by the other B1 haplotypes? It is not clear in the text but I've assumed they were trimmed out by this selection criteria. In this way, an important fraction of the ITS2 sequence richness might be hidden (148 OTUs?) and this criteria must either be re-evaluated or clarified.

The stringency of this selection and the duplication levels observed for the ITS2 sequence in the literature (Thornhill et al (2007) has found 13 ITS2 clones on a B1 Symbiodinium clonal culture, 5 non-functional ones) favors the interpretation that these haplotypes are non-functional pseudogenes, intragenomic variants of the haplotype II (line 332). Also, secondary structure analysis could be done in order to evaluate whether this haplotypes are functional or not.

Additional comments

The article makes a clear report on the use of Next-generation sequencing (NGS) for studying the Symbiodinium diversity in an interesting environment and is generally well writen. It's an important report an upcoming technology that shall be used more frequently and I believe it should be ready for publishing after clarifying the points mentioned above.

The articles correctly points out the caveats of deep amplicon sequencing but I think it can benefit from a discussion on the advantages of NGS over the prevailing methodologies used for Symbiodinium diversity assessment.
Besides being more efficient (in terms of time, number of sequences and cost per base), NGS has some technical advantages over DGGE and bacterial cloning, currently prevailing methodologies. This advantages were already observed in other microbial groups, but are not discussed neither in the the manuscript nor in the listed references. Briefly, it is less vulnerable to replication errors than bacterial cloning (in the sense that sequences from mixed clusters will be discarded) and might retrieve a higher sequence diversity than DGGE (when only the dominant band is sequenced). The two references below adds up to the methodology debate and could be included in the manuscript:

LaJeunesse TC, Thornhill DJ (2011) Improved resolution of reef-coral endosymbiont (Symbiodinium) species diversity, ecology, and evolution through psbA non-coding region genotyping. PLoS ONE 6(12): e29013. doi:10.1371/journal.pone.0029013

Stat M, Bird CE, Pochon X, Chasqui L, Chauka LJ, et al. (2011) Variation in Symbiodinium ITS2 sequence assemblages among coral colonies. PLoS ONE 6(1): e15854. doi:10.1371/journal.pone.0015854

·

Basic reporting

- The article is written in standard, professional, clear English.
- The article is well referenced.
- The article has a standard template.
- Figures/Tables are relevant.
- The submission is self-contained, but the discussion spreads a bit further than is relevant to the dataset.

Experimental design

- This is original primary research, with a clearly defined research question, performed to a high technical standard.
- There are some issues of reproducibility of the analysis (i.e. thresholds for clustering) that I detail in my long response below.
- I see not ethical issues.

Validity of the findings

- Data are generally well analyzed statistically.
- I'm not clear on how/whether/in what form they released the raw sequence data.
- The conclusions are generally well-supported (with the exception of the weak structure I comment on below)
- While I won't ding them for too much speculations, there are areas in the discussion that aren't truly relevant to _this_ dataset.
- While the results may be of interest to specific researchers, I suggest a few analyses to broader the appeal of the work, and quantitatively support descriptive caveats the authors make in the discussion.

Additional comments

Here is my general "standard" review format:

This paper, “Next-generation sequencing reveals cryptic Symbiodinium diversity within Orbicella faveolata and Orbicella franksi at the Flower Garden Banks, Gulf of Mexico”, is a well-referenced, competently analyzed study that describes distinctions among the Symbiodinium communities of two coral species from two deeper water (~20m) sites, separated by 19 km.

Using coral host microsatellite (9 loci), and 454 amplicon Symbiodinium ITS2, individually barcoded samples, the authors describe host population genetic distinctions among species and sites, as well as distinctions in the relative frequencies of ITS2 haplotypes among species and sites. The authors claim that STRUCTURE reveals distinctions among species, but not among sites, and that AMOVA shows no significant structure among either species or sites. The authors then cluster the ITS2 sequences, focusing only on those 5 OTUs (of 153) that were present in more than half of all samples, and show that two of these show significant distinctions among sites for both species, and a site:genotype:species interaction term.

Generally speaking, the paper is very well referenced and the intro and discussion presents a deep understanding of the relevant issues in the field. Also, the analyses the analyses are preformed competently, and for the most part presented accurately and with due caution.

However, I have some main problems with the paper, and a few more minor issues: The big ones are first an issue of (1) too much presented in the discussion that’s not relevant to this dataset, and (2) a few missed opportunities to tackle the fundamental problems presented by both amplicon sequencing and ITS2 as a marker.

I would recommend publication after a major revision to address these two issues, and a set of smaller problems.

Big Stuff:
To start with (1): the discussion takes great steps to describe why this sort of work is important by discussing issues related to symbiont shuffling/switching and mesophotic reefs. From my perspective, this dataset doesn’t speak well to Symbiont Plasticity, or the importance of studying mesophotic reefs. On plasticity, deep-amplicon sequencing can address these issues well, but you either need a time-series dataset or very distinct environments (likely with a transplant), neither of which exist in this study. Also, the FGB areas the study focuses on are only at ~20m, which most researchers wouldn’t count as truly mesophotic depths. So, while these issues are important to the field, I don’t find them really relevant to this paper’s discussion.

However, in the next two sections of the discussion, the authors raise really important issues with using deep amplicon sequencing, that their dataset has the potential to make substantive contributions to the understanding of, but they just discuss the issues without hitting the nitty-gritty analysis they could. There are two major issues here – one is the challenge of distinguishing low-frequency functional alleles from pseudogenes and sequencing error, and the second is dealing with intra-genomic variation:

Low-Frequency Alleles: This is a fundamental problem with deep-sequence data, and the authors document that their low-amplification approach, and conservative selection of common alleles avoid some issues presented by sequencing error. However, their documentation of the low-divergence, low-frequency variants they find, how they chose the clustering level, or how one should decide where this balance between the real-and-rare vs. spurious lies could be much better developed to generate more confidence in their approach. The OTU clustering steps are no-where near as clear as they could be, e.g. how exactly did the authors get from 20K+ unique sequences to 153 OTU clusters?

Psuedogenes: The authors openly admit that the haplotypes they follow may be psuedogenes, without taking any opportunity to test whether this may be true by mapping the SNPs to ITS2 structure (a la Thornhill, LaJeunesse, Santos 2007; 4SALE). Especially for the 5 you select, this should be trivial. The shorter length of these sequences (290bp) may make this more challenging than longer reads, but at least describing efforts in this line would increase the reader’s confidence that the authors have dealt well with these issues.

Intra-genomic Variation: Finally, given the fact that individual samples have been uniquely barcoded, the authors should be in an enviable position to document intra-genomic variation by showing how ITS2 variants map across individual, species and site. ITS2 is widely relied upon as the canonical marker for Symbiodinium systematics, though all involved know that it has issues with intra-genomic variation. Documenting that variation in this apparently low-diversity population would be very useful, especially if coupled with the psuedogene folding algorithms mentioned above.

To be clear, I don’t feel that the authors need to completely restructure their analysis to focus on this issue, but I think it presents an opportunity to provide a paper that would generate broader interest for the community. Something as easy as partitioning the variation for the 153 OTUs into within individual, species and site would give the reader another reason to come to this paper, beyond a general interest in deeper-water Orbicella communities.

Smaller stuff:
Abstract: “both coral species contained only Symbiodinium identifiable as clade B type B1” This is not strictly true – three haplotypes had major (10+ bp insertion/deletions from B1). “most closely related to type B1” would be more accurate.
Otherwise, abstract is fine, but may require revision after refocusing the rest of the manuscript.
Introduction: While I understand how hard it is to write passages about the symbiotic relationship during bleaching, something about describing a symbiosis undergoing bleaching as a “severely broken symbiosis” strikes me as over-dramatic and potentially inaccurate. Try rewriting that section.
When discussing clade-level physiological distinctions:
This needs to be much more carefully worded to describe the difficulty of ascribing physiological distinctions to genotypic distinctions in Symbiodinium. Many of the earlier clade-level physiological distinctions were made at the resolution of the genetic markers, not due to an accurate representation of a relevant phenotypic distinction. While the literature does back you up on these points, the physiological changes are almost certainly more granular that you propose. I believe the authors understand this point, but just need to take more care to ensure that their readers are not confused on this point.
Methods:
“New 30 μL PCR reactions were performed to attach A and B Rapid adaptors specific for
454 GS FLX.” – In this section, the authors should be clearer that they have individual sample barcodes.
“The clustering algorithm usearch was used to cluster reads into operational taxonomic units (OTUs) (Edgar 2010).” You need to be clearer about what settings you used for this clustering, and what they mean – this step gets to the heart of what OTUs you consider “real”, and shouldn’t be glossed over.

Results:

“This result was confirmed by AMOVA analysis (Table 1). AMOVA results comparing FST between species and sites showed no significant genetic differentiation between the two host species collected at each site (Table 1).”

This appears not to be true, given the P-values shown in table 1.
Aside from the cross-species comparison, O. faveolata seems to show more consistent distinctions in the structure analysis (K=3, with the 3rd cluster restricted to one bank), Fst (low but significant Fst), and more consistent distinctions in Haplotypes IV and V (than O. franksi).

“After removing all reads shorter than 290bp, 122,867 reads representing
20,260 unique sequences remained. Clustering the unique sequences yielded 153 OTUs. Mapping the original filtered reads to these 153 OTUs revealed that only five of the OTUs were detected in more than half of all coral individuals sequenced.”

Again, we need to know the clustering algorithm's settings, i.e. what level of diversity existed within-cluster, and why those “detected in more than half of all samples” represents the right balance between being conservative to avoid seq. error and now ignoring real data.

Discussion:
“As a consequence, we hypothesize the monotypic Symbiodinium species seen at the FGB for O. faveolata and O. franksi do not show more diverse populations because of the lack of genetic divergence at the host level and the similar environmental conditions at both banks.”

You can make this claim more interesting by starting with this conclusion (high gene flow, similar environments) but then pointing out the weak, parallel structure in both hosts and symbiont communities. (See comment above about Fst values).

Potential roles of mesophotic reefs,
This paragraph is arguably irrelevant to your actual dataset. You could do a better job of arguing that in the turbid gulf these depths are truly mesophotic, and compare your results to areas that are similar but have higher light environments (from the literature). As is, I don’t buy it. Also:

“The pristine and undisturbed conditions at the FGB” I don’t really buy that either: “relatively undisturbed” I could accept.

Plasticity of symbiosis
Again, this comes off as not really relevant to your dataset.

Using deep amplicon sequencing to detect species diversity:
“By annealing unique barcodes to” – given that you’ve got samples with unique barcodes, you should really be able to better address intra-genomic varation.

“Illumina” You switch platforms half-way through the discussion with no guidance. If you’re going to look to the future on a different platform, you need to discuss it better.

“these haplotypes might be prospective pseudogenes” You even cite the relevant paper that should guide the folding analysis using 4SALE (Siebel et al. 2006, 2008; http://4sale.bioapps.biozentrum.uni-wuerzburg.de/).

“Given unknown whole and partial genome duplication events in Symbiodinium some of these reference haplotypes could potentially come from the same genome (Hou & Lin 2009). Empirical analyses may predict copy numbers but do not provide conclusive results for inter versus intra-genomic haplotypes.” That said, you could do a much better job of addressing this issue, and it would greatly increase the appeal of the paper.


I hope you found this review constructive and fair.

Tom Oliver

---

## Round 0.2 · accepted · Accept

Congratulations. The paper is now accepted.